# Chronic oxytocin administration stimulates the oxytocinergic system in children with autism

Matthijs Moerkerke [1,2,6], Nicky Daniels [2,3,6], Laura Tibermont[1,2], Tiffany Tang[1,2], Margaux Evenepoel [2,3], Stephanie Van der Donck[1,2], Edward Debbaut [1,2], Jellina Prinsen [2,3], Viktoria Chubar[4], Stephan Claes[4], Bart Vanaudenaerde[5], Lynn Willems[5], Jean Steyaert[1,2], Bart Boets [1,2,7] & Kaat Alaerts [2,3,7] ✉

Clinical efficacy of intranasal administration of oxytocin is increasingly explored in autism spectrum disorder, but to date, the biological effects of chronic administration regimes on endogenous oxytocinergic function are largely unknown. Here exploratory biological assessments from a completed randomized, placebo-controlled trial showed that children with autism (n = 79, 16 females) receiving intranasal oxytocin for four weeks (12 IU, twice daily) displayed significantly higher salivary oxytocin levels 24 hours after the last oxytocin nasal spray administration, but no longer at a four-week follow up session. Regarding salivary oxytocin receptor gene (*OXTR*) epigenetics (DNA-methylation), oxytocin-induced reductions in *OXTR* DNA-methylation were observed, suggesting a facilitation of oxytocin receptor expression in the oxytocin compared to the placebo group. Notably, heightened oxytocin levels post-treatment were significantly associated with reduced *OXTR* DNA-methylation and improved feelings of secure attachment. These findings indicate that four weeks of chronic oxytocin administration stimulated the endogenous oxytocinergic system in children with autism.

Autism spectrum disorder is a neurodevelopmental condition, characterized by repetitive and restrictive behaviours and socio-communicative difficulties for which limited therapeutic options exist to date[1]. In the past decade, intranasal administration of oxytocin has been increasingly explored as a new approach to facilitate social development and reduce disability associated with autism[2]. To date, two main mechanistic frameworks on oxytocin's role in regulating social behaviour have been proposed. First, the anxiolytic account suggests that oxytocin primarily regulates stress and social anxiety responses, thereby promoting social approach behaviour during interactions[3–6]. Secondly, the social salience hypothesis proposes that

oxytocin enhances attention to and perception of social cues by prioritizing neural resources for processing these cues[7]. However, while initial single-dose administration studies yielded promising acute effects of oxytocin on pro-social behaviour[2], subsequent multiple-dose, chronic administration studies (i.e. administrating the oxytocin nasal spray over a course of multiple weeks) have yielded a more mixed pattern of results, with some studies demonstrating beneficial outcomes, while others did not[8–12] (see[13] for a full review on chronic oxytocin administration trials).

In light of the growing number of studies demonstrating a limited bioavailability of naturally occurring oxytocin in autistic children[14–16], it

[1]Center for Developmental Psychiatry, Department of Neurosciences, KU Leuven, Leuven, Belgium. [2]Leuven Autism Research (LAuRes), KU Leuven, Leuven, Belgium. [3]Research Group for Neurorehabilitation, Department of Rehabilitation Sciences, KU Leuven, Leuven, Belgium. [4]University Psychiatric Centre, KU Leuven, Leuven, Belgium. [5]Laboratory of Respiratory Diseases and Thoracic Surgery, Department of Chronic Illness and Metabolism, KU Leuven, Leuven, Belgium. [6]These authors contributed equally: Matthijs Moerkerke, Nicky Daniels. [7]These authors jointly supervised this work: Bart Boets, Kaat Alaerts. ✉e-mail: kaat.alaerts@kuleuven.be

is crucial to gain deeper insights into how exogenous administration can impact the functioning of the endogenous oxytocin system. This is important, considering that biological changes likely underlie oxytocin-induced behavioural-clinical outcomes[17] and therefore allow delineating possible biological mechanisms of inter-individual variation in clinical treatment responses.

To date, insights into how exogenous administration impacts endogenous oxytocinergic signalling have predominantly emerged from single-dose administration studies, showing elevated salivary oxytocin levels up to 2 and even 7 hours after administration in autistic and non-autistic populations[18–24]. Considering that the half-life of oxytocin is only a few minutes in blood plasma and 20 minutes in cerebrospinal fluid[25], sustained high levels of salivary oxytocin likely reflect an upregulation of endogenous oxytocin release induced by its acute exogenous administration. This notion is supported by a recent chronic administration study, examining the effect of a four-week course of daily oxytocin administrations on salivary oxytocin levels in autistic adult men[26]. Here, elevated salivary oxytocin levels were shown up to four weeks after cessation of the nasal spray administration period, indicating a self-perpetuating elevation of oxytocin levels through a feed-forward triggering of its own release, in line with the notion of a 'positive spiral of oxytocin release' as suggested before by De Dreu (2012)[27]. More research is needed, however, to understand the impact of chronic oxytocin administration on its endogenous production, especially in autistic children, considering that oxytocin can preferably exert its therapeutic potential within early life developmental windows.

Aside assessments of circulating oxytocin, also variations in (epi)genetic modifications of the oxytocin receptor gene (*OXTR*) are considered important markers of endogenous oxytocinergic function. DNA methylation (DNAm) is one of the most extensively studied epigenetic mechanisms involved in the regulation of gene transcription, with increased DNAm frequency found to be associated to decreased gene transcription, and therefore less receptor expression and availability[28]. While initial studies have linked increased *OXTR* DNAm to social difficulties and traits associated with autism[29], insight in whether (chronic) oxytocin administration impacts *OXTR* DNAm and whether these changes relate to altered oxytocin levels remains currently unexplored.

Here exploratory biological assessments from a completed double-blind, randomized, placebo-controlled study were performed to examine the effects of a four-week course of chronic oxytocin administration (12 IU, twice daily), or placebo, on oxytocinergic function as assessed using salivary oxytocin levels and *OXTR* DNAm in school-aged children with autism. To do so, salivary oxytocin and DNA samples were collected at baseline, prior to nasal spray administration (T0); immediately after the four-week administration period, at least 24 hours after the last nasal spray administration (T1); and at a follow-up session, four weeks after cessation of the daily administrations (T2).

Importantly, the exploratory biological salivary collections were part of a larger protocol, additionally including the assessment of oxytocin-induced changes on clinical-behavioural scales, as outlined in more detail in Daniels et al. (2023)[12]. Overall, Daniels et al. [12] showed that - compared to children receiving placebo - the group of children receiving oxytocin did not display stronger improvements in parent-reported social functioning (assessed as primary behavioural outcome using the Social Responsiveness Scale) or repetitive behaviours, attachment or anxiety, although the combination of oxytocin administration with psychosocial treatment might suggest a synergetic effect[12]. To understand possible inter-individual variations in clinical treatment outcomes, we here explored whether individual variation in oxytocin-induced effects on endogenous oxytocinergic function may relate to variation in clinical effects in terms of social function, repetitive behaviours, attachment and anxiety.

## Results

Children with autism participating in this double-blind, randomized, placebo-controlled study, were randomly allocated to receive four weeks of intranasal oxytocin administration (12IU twice daily) or placebo (see Fig. 1 for the CONSORT flow diagram visualizing the number of participants randomized and Table 1 for the number of participants analyzed).

### Effect of chronic oxytocin administration on salivary oxytocin levels

Exploratory assessments of salivary oxytocin levels were acquired before (T0) and after the four-week nasal spray administration period (post: T1, follow-up: T2). At each assessment, both a morning and an afternoon sample were acquired.

**Morning oxytocin levels.** Mixed-effect analyses with the fixed factors 'nasal spray' (oxytocin, placebo) and 'assessment session' (T1, T2) (and the inclusion of baseline T0 scores as covariate) revealed a significant main effect of 'nasal spray' ($F(1,74) = 15.50$; $p < 0.001$; $\eta_p^2 = 0.18$) and 'assessment session' ($F(1,74) = 10.15$; $p = .002$; $\eta_p^2 = .13$). Yet, these main effects need to be interpreted in the context of the significant 'nasal spray x assessment session' interaction effect, indicating that morning oxytocin levels were significantly augmented in the oxytocin group, compared to the placebo group ($F(1,74) = 12.19$; $p < 0.001$; $\eta_p^2 = 0.15$), but only at the T1 postassessment ($p_{Bonferroni} < 0.001$), not at the T2 four-week follow-up ($p_{Bonferroni} > 0.05$), see Fig. 2A.

**Afternoon oxytocin levels.** An overall similar pattern of results was identified for afternoon oxytocin levels, indicating a significant augmentation in the oxytocin group compared to the placebo group ($F(1,71) = 7.06$; $p = 0.010$; $\eta_p^2 = 0.09$), at the T1 post assessment ($p_{Bonferroni} = 0.001$), not at the T2 follow-up ($p > 0.05$), see Fig. 2B.

### Effect of chronic oxytocin administration on salivary *OXTR* DNAm

Epigenetic salivary samplings were performed at each assessment session (T0, T1, T2) and variations in *OXTR* DNAm were assessed at three CpG sites of the *OXTR* gene, previously shown to be impacted in autism (i.e., -934, −924 and -914, see[29]).

**CpG site −924.** A significant main effect of *nasal spray* was identified ($F(1,70) = 7.76$; $p = 0.007$; $\eta_p^2 = 0.10$), indicating that DNAm of CpG site −924 was significantly reduced in the oxytocin group, compared to the placebo group, see Fig. 2C. The absence of a *spray x assessment session* interaction ($F(1,70) = 0.90$; $p = 0.346$; $\eta_p^2 = 0.01$) indicates that the main effect of nasal spray was evident across both the T1 post and T2 follow-up assessment. The significant main effect of *assessment session* indicates overall higher DNAm at the T2 assessment compared to the T1 assessment ($F(1,70) = 6.76$; $p = 0.011$; $\eta_p^2 = 0.085$).

**CpG site -914 and -934.** No significant main nor interaction effects were identified for CpG site -914 and -934 (all $p > 0.05$), indicating no differential modulation of DNAm at these sites after oxytocin or placebo nasal spray administration, see Fig. S1.

### Associations of oxytocin levels with *OXTR* DNAm

Notably, in the oxytocin group, a significant association was evident between salivary oxytocin levels post-nasal spray, at T1, and DNAm of CpG site −924 ($\rho = -0.39$; $p = 0.018$), indicating that children of the oxytocin group displaying higher oxytocin levels at T1 also showed a stronger reduction in DNAm of CpG site -924 (reflecting higher oxytocin-receptor expression), see Fig. 2D. Note that the relationship was only evident at the T1 assessment session - at which treatment-induced elevations in salivary oxytocin levels were observed – but no longer at the T2 follow-up session ($\rho = -0.21$; $p = 0.214$). Also no

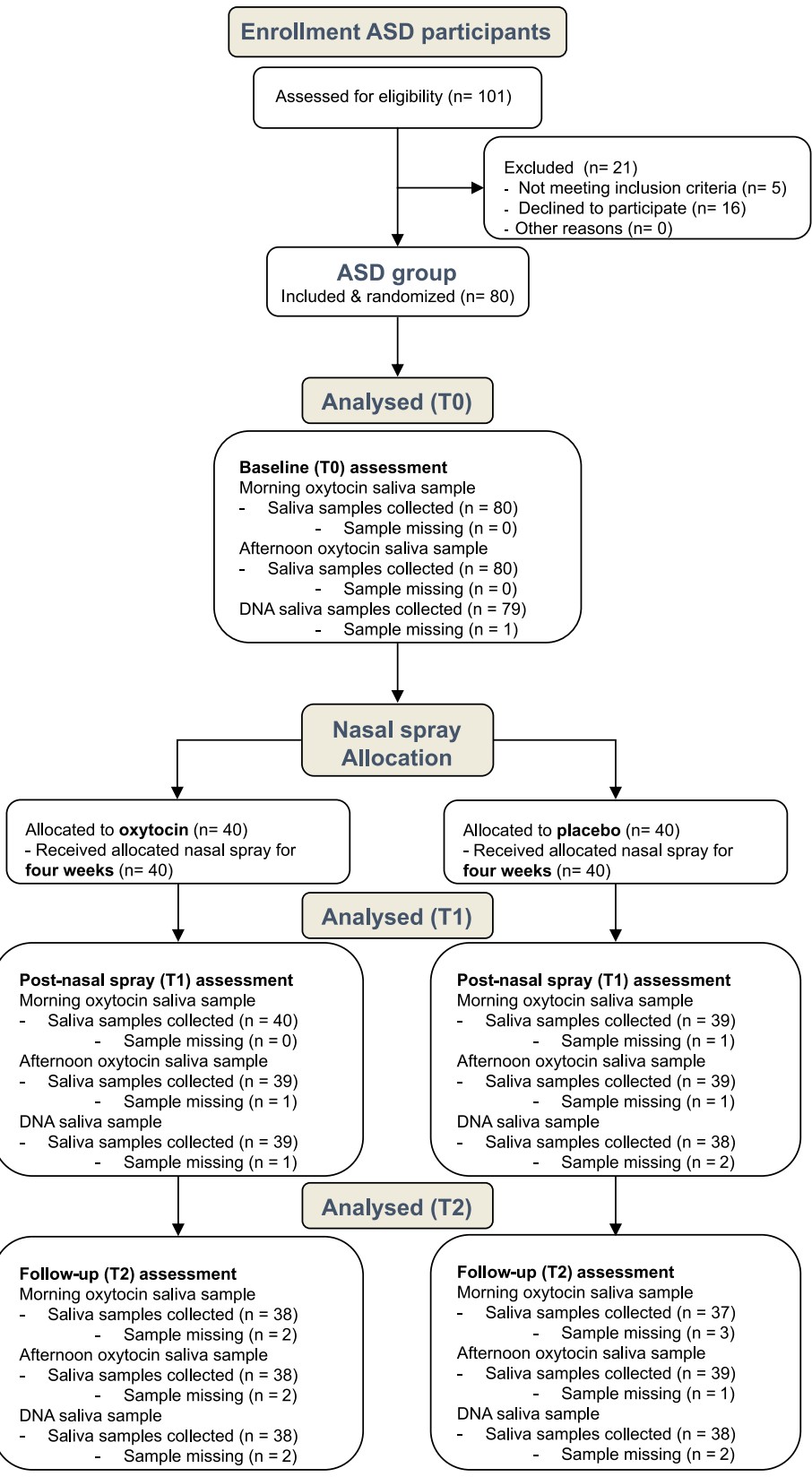

**Fig. 1 | CONSORT flow diagram.** Participants were assessed for eligibility prior to the baseline session (T0) and after the session allocated to receive either oxytocin or placebo nasal sprays for a four week period of twice daily nasal spray administration. Saliva oxytocin (morning and afternoon) and DNA samples were collected at baseline (T0), immediately after the four-week administration period (T1; at least 24 hours after the last nasal spray administration), and at a follow-up session, four weeks after cessation of the daily administrations (T2). As outlined, for some participants, saliva samples were missing at one or more assessment sessions due to participant being unable or forgetful to bring the sample or due to discontinuation of the study by the participant.

## Table 1 | Demographic and clinical characteristics of children randomized to receive oxytocin or placebo

| | Oxytocin group | | Placebo group | | | |
| --- | --- | --- | --- | --- | --- | --- |
| | N | Mean ± SD | N | Mean ± SD | t value | p value |
| **Age** | 40 | 9.9 ± 1.3 | 39 | 9.8 ± 1.2 | 0.55 | 0.59 |
| **WISC-V[a]** | | | | | | |
| Verbal IQ | 39 | 105.6 ± 14.5 | 38 | 109.3 ± 15.7 | -1.07 | 0.29 |
| Performance IQ | 40 | 103.0 ± 15.4 | 38 | 101.5 ± 12.7 | 0.45 | 0.65 |
| **Biological sex** | | | | | | |
| Female | 8 (20%) | | 8 (21%) | | | |
| Male | 32 (80%) | | 31 (79%) | | | |
| **Handedness** | | | | | | |
| Left | 4 (10%) | | 6 (15%) | | | |
| Right | 36 (90%) | | 33 (85%) | | | |
| **ADOS-2[b]** | | | | | | |
| Social affect | 33 | 7.4 ± 3.7 | 32 | 7.5 ± 3.7 | -0.05 | 0.96 |
| Restricted and repetitive behavior | 33 | 2.1 ± 1.2 | 31 | 1.7 ± 1.3 | 1.34 | 0.19 |
| Total | 35 | 9.8 ± 3.9 | 32 | 9.2 ± 4.2 | 0.63 | 0.53 |
| **SRS-2[c]** | | | | | | |
| Total | 40 | 90.1 ± 22.8 | 39 | 87.9 ± 20.0 | 0.46 | 0.65 |
| | | | | | | Pearson chi-square |
| **Psychostimulant medication use** | 11 | | 9 | | | 0.20 | 0.90 |
| **Psychiatric comorbidity** | 11 | | 7 | | | 0.89 | 0.64 |

Mean baseline scores, from participants completing at least one post-nasal spray session, are listed separately for each nasal spray administration group. T- and p-values correspond to independent sample t-tests assessing between-group differences in baseline scores.
[a]Wechsler Intelligence Scale for Children, WISC-V-NL[49]. The verbal intelligence quotient (IQ) was derived from the subtests Similarities and Vocabulary. The performance IQ was derived from the subtests Block design and Figure puzzles
[b]Autism Diagnostic Observation Schedule, 2nd edition, ADOS-2, module 3[47]
[c]Social Responsiveness Scale, 2nd edition, SRS-2[48]. Significant difference at $p < 0.05$ statistical threshold.

significant association was evident at the T0 session ($\rho = 0.15$; $p = 0.371$), indicating no significant relationship between baseline oxytocin levels and baseline DNAm of CpG site -924.

## Associations with behavioural-clinical reports

A significant relationship was identified between salivary oxytocin levels assessed from the oxytocin group at T1 (at which treatment-induced elevations were evident) and self-reports of secure attachment towards peers (ASCQ), indicating that children of the oxytocin group displaying higher oxytocin levels at T1 also showed higher reports of secure attachment at T1 ($\rho = 0.35$; $p = 0.030$), see Fig. 3A). Moderate (but nonsignificant) opposite relationships were evident for self-reported attachment avoidance ($\rho = -0.30$; $p = 0.070$, Fig. 3B), parent-reports of social difficulties (SRS-2; $\rho = -0.31$; $p = 0.062$, Fig. 3C) and anxiety (SCARED-NL; $\rho = -0.32$; $p = 0.053$, Fig. 3D) all indicating that improved clinical presentations at T1 were associated with higher T1 oxytocin levels. No significant associations were evident between oxytocin levels and clinical-behavioural scales at the follow-up session T2 (all $p > 0.208$), or at the baseline T0 assessment (all $p > 0.170$).

Also no significant associations were identified between levels of DNAm of CpG site -924 and the abovementioned clinical scales, either at T1, T2 or T0 (all $p > 0.108$).

## Discussion

This study examined the impact of a four-week chronic oxytocin administration regime on endogenous oxytocinergic function in autistic children. Salivary oxytocin levels were reliably increased 24 hours after the last oxytocin nasal spray, but no longer at the follow-up session, four weeks after cessation of the oxytocin administrations. Reduced *OXTR* DNAm levels were also observed, suggesting a facilitation in oxytocin-receptor expression, as previously evidenced[28,30], in the oxytocin, compared to the placebo group, up to four weeks after the last nasal spray administration. Increased oxytocin levels were significantly associated with reduced *OXTR* DNAm levels, as well as with improved clinical presentations on self-reported feelings of secure peer attachment. These findings provide evidence that the chronic exogenous administration induced stimulation of endogenous oxytocinergic system in children with autism.

Although the exact physiological mechanisms remain unclear, prior research showed that oxytocin-receptor binding promotes secretion of additional oxytocin and affects DNA transcriptional activation/repression[25,31,32]. Particularly, an auto-excitatory mechanism of somato-dendritic release of oxytocin from hypothalamic cells has been described, indicating that auto-binding of oxytocin to its G protein-coupled receptors is sufficient to elevate intracellular $Ca^{2+}$ levels from internal storage, in turn triggering a feed-forward of dendritic oxytocin exocytosis (i.e. release)[25,33].

With regard to the regulation of *OXTR* expression and receptor availability, prior rodent research showed mixed results, with some studies showing a receptor downregulation upon chronic application of oxytocin in mice (10 ng/h, over a period of 14 days)[34,35], while others showed opposite effects[36]. For example, in oxytocin knockout mice, the expression of the *OXTR* gene was shown to be reduced in hippocampal tissue, due to unavailability of the ligand[36]. In the current human sample, the chronic oxytocin administration induced a predominant decrease in *OXTR* methylation, previously shown to be reflective of increased receptor expression[28,30]. It is likely, however, that the effects on gene regulation of (chronic) oxytocin administration are dependent on the administered dose and frequency, with some dosing schemes inducing facilitation, while others might induce downregulation of receptor availability. In a recent study by Le et al. (2022), the possibility of an intermittent dosing scheme - administering oxytocin intranasally every other day for a period of 6 weeks to avoid pharmacological overstimulation - was explored in a paediatric sample of young autistic children, and although no direct assessments of epigenetic changes were reported, the administration regime yielded strong beneficial effects on various clinical-behavioural outcomes assessing social function[10]. It should be interesting for future studies to further explore optimal dosing schema's for yielding maximal endogenous stimulation of the oxytocinergic system. Since epigenetic mechanisms constitute a direct pathway by which environmental changes (e.g. increased oxytocin availability) may impact DNA transcriptional activation/repression, the observed changes in DNAm of *OXTR* likely reflect an important biological pathway through which chronic oxytocin administration regimes may induce long-lasting neurobiological and possibly associated clinical-behavioural effects. In short, the elevated oxytocin availability (through its exogenous administration) may have induced a feed-forward mechanism of the endogenous oxytocinergic function by increasing its own release, as well as prompting more sustained epigenetic *OXTR* modifications. The stimulating effect of chronic oxytocin administration on peripheral oxytocin levels in autistic children, observed at T1 but not at T2, is partly in line with prior observations in adults with autism[26]. In the latter study, increases in endogenous oxytocin levels persisted up to four weeks after cessation of the chronic nasal spray administrations. Previously, developmental differences of the oxytocinergic system have been evidenced, indicating lower endogenous oxytocin levels in autistic children, but not adults[14]. Perhaps in autistic children, more so

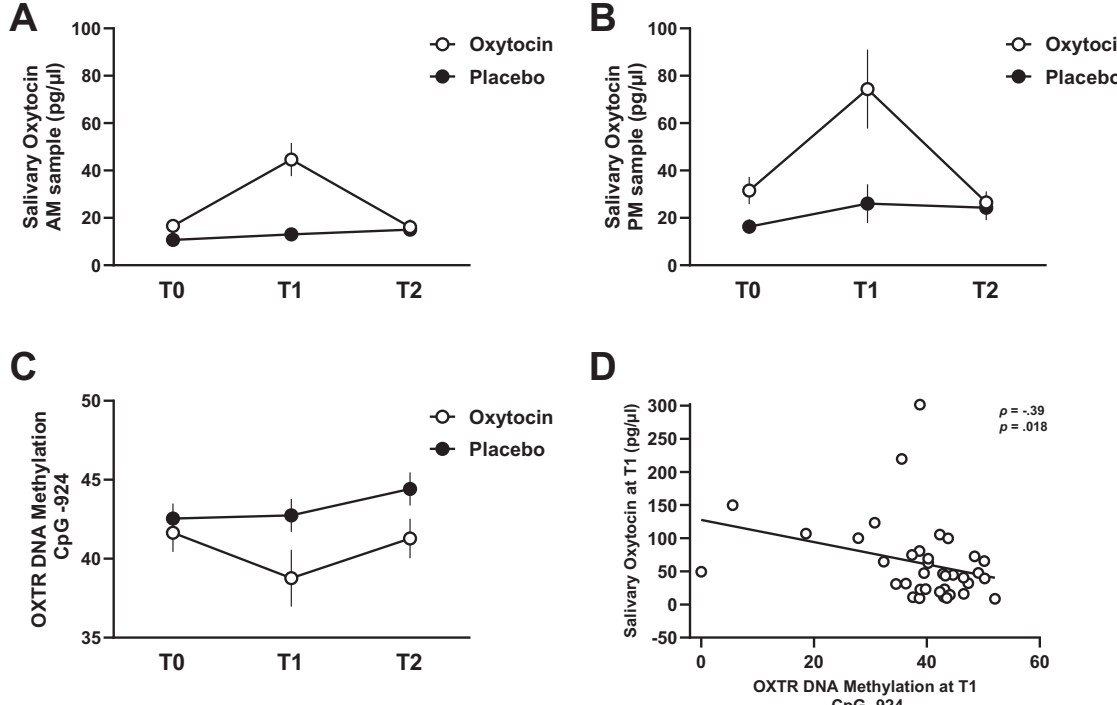

**Fig. 2 | Effect of chronic oxytocin administration on salivary oxytocin levels and *OXTR* DNAm levels.** Salivary oxytocin levels are visualised for each nasal spray group (oxytocin and placebo) at each assessment session: baseline (T0), ≥24 hours post-nasal spray (T1) and four weeks post-nasal spray (T2). Panel **A** shows the morning oxytocin levels (AM sample; T0: $n_{oxytocin}$ = 39, $n_{placebo}$ = 39; T1: $n_{oxytocin}$ = 39, $n_{placebo}$ = 36; T2: $n_{oxytocin}$ = 38, $n_{placebo}$ = 34) and panel **B** the afternoon oxytocin levels (PM sample; T0: $n_{oxytocin}$ = 38, $n_{placebo}$ = 39; T1: $n_{oxytocin}$ = 39, $n_{placebo}$ = 39; T2: $n_{oxytocin}$ = 38, $n_{placebo}$ = 37). Panel **C** visualises the salivary *OXTR*

DNAm levels at CpG −924 for each nasal spray group (oxytocin and placebo) at each assessment session (T0: $n_{oxytocin}$ = 38, $n_{placebo}$ = 39; T1: $n_{oxytocin}$ = 37, n$_{placebo}$ = 38; T2: n$_{oxytocin}$ = 38, n$_{placebo}$ = 37). Data are presented as mean values, vertical bars denote standard errors. Panel **D** visualises the association of salivary oxytocin levels (averaged over the morning and afternoon samples) with the salivary *OXTR* DNAm levels in CpG −924 in the oxytocin group at T1. Source data are provided as a Supplementary Data Source file.

than in adults, homeostatic responses upon cessation of the chronic oxytocin administration occur faster, thereby reducing heightened levels of circulating oxytocin more rapidly to initial baseline levels. Alternatively, adults (as compared to children) might adapt their social interactive behaviour more readily upon receiving oxytocin administrations, i.e., through increasingly engaging in and positively experiencing of socially stimulating contexts. According to the positive spiral of oxytocin release, this increase in (positive) social experiences may be pivotal for further facilitating oxytocin's endogenous release, also in the long-term, as observed up to four-weeks after cessation of the nasal spray administration regime in adults[26], but not children in the current cohort. In light of potential treatment applications, these observations highlight the importance of combining oxytocin administration with social stimulation (e.g. psychosocial therapy) to perpetuate the endogenous oxytocin feedforward-loop[27]. This aligns with emerging evidence indicating that oxytocin administration as stand-alone treatment may be suboptimal[37,38], and that combinatory approaches, combining the chronic oxytocin administrations to concomitant psychosocial training interventions will be pivotal for maximizing its full therapeutic potential and clinical efficacy[10,12].

It is noteworthy that higher levels of endogenous oxytocin observed post-treatment were linked to better self-reported feelings of secure attachment. Additionally, there was a trend towards a reduced self-reported attachment avoidance, as well as parent-reported social difficulties and anxiety. A large body of literature links variations in circulating oxytocin to variations in behavioural manifestations, including prosociality and attachment-related constructs[39–41]. Over the past decades, the role of the oxytocinergic system in inter-personal bonding and formation and maintenance of secure attachment bonds was highlighted. Our study provides evidence of the link between

elevated oxytocin levels and a better clinical presentation of secure attachment in autistic children, supporting the notion that also in autistic children heightened oxytocin may facilitate a sense of (social) security and safety[42,43].

In the current study, salivary sampling was adopted for both oxytocin level and *OXTR* DNAm characterizations, as these assessments were most feasible in a paediatric population. While correlations have been demonstrated between salivary and central (cerebrospinal fluid) oxytocin levels[44], salivary oxytocin may constitute only a proxy of central oxytocinergic function. Future studies are therefore warranted to examine whether the currently reported effects - both in terms of salivary oxytocin and epigenetic modification – would hold also for other biological material, including blood and cerebrospinal fluid. Moreover, the *OXTR* DNAm results were only found at CpG site -924, even though all three investigated sites fall within the same region in the CpG island of the *OXTR*, shown to regulate the expression of the oxytocin receptor[28]. Future studies should aim to clarify this differential effect, and specifically, how altered methylation at CpG site -924 might have impacted oxytocin receptor gene expression. Further, considering the tight age range of our school-aged children with autism, and a predominant sampling from autistic boys, it should be explored whether the observed effects will generalize across different age ranges and in samples with a larger representation of girls.

To conclude, four weeks of chronic oxytocin administration was shown to stimulate the endogenous oxytocinergic system in autistic children, as evidenced by increased salivary oxytocin levels and associated decreased *OXTR* DNAm levels (previously shown to reflect increased oxytocin-receptor expression[28,30]). Furthermore, elevated oxytocin levels post-treatment were associated with a

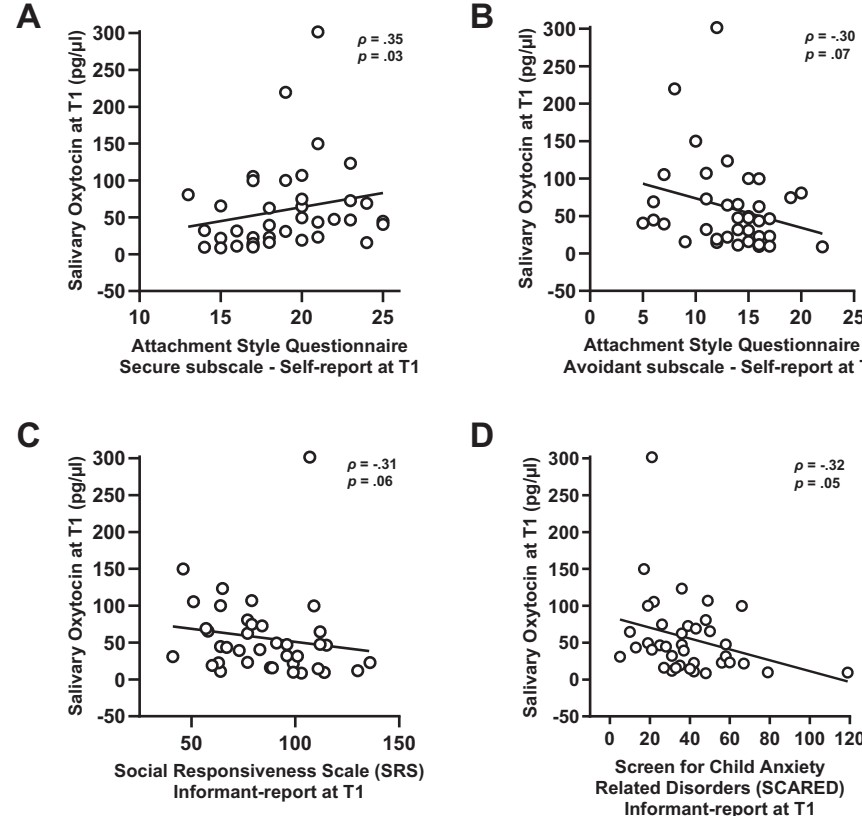

**Fig. 3 | Associations of oxytocin levels with questionnaires assessing attachment style, social responsivity and anxiety.** The association of salivary oxytocin levels with questionnaire scores at the postoxytocin nasal sprays session (T1) are visualised. Panel **A** shows the association with secure-attachment (Attachment Style Classification Questionnaire, ASCQ). Panel **B** shows the association with avoidant-attachment (ASCQ). Panel **C** shows the association with social responsiveness (Social Responsiveness Scale, SRS). Panel **D** shows the association with anxiety scores (Screen for Child Anxiety Related Emotional Disorders, SCARED). Source data are provided as a Supplementary Data Source file.

better clinical presentation, indicating an important mechanistic link between oxytocin's chronic biological and clinical-behavioural effects.

## Methods

All study procedures and consent forms were approved by the local Ethics Committee at the KU Leuven (S61358) in accordance with Declaration of Helsinki. Parents/legal guardians of the participants provided informed consents and participants provided informed assents, prior to the start of the study. The trial was registered at the European Clinical Trial Registry (EudraCT 2018-000769-35) and the Belgian Federal Agency for Medicines and Health products (see Supplementary methods).

### Study design

A double-blind, randomized, placebo-controlled study with parallel design was conducted at the Leuven University Hospital (Belgium) to assess the effects of four weeks of intranasal oxytocin administration (12IU twice daily) on endogenous levels of salivary oxytocin and *OXTR* DNAm in school-aged children with autism (see Fig. 1 for the CONSORT flow diagram visualizing the number of biological samples collected and analysed for each assessment session). Saliva oxytocin and DNA samples were collected at baseline, prior to nasal spray administration (T0); immediately after the four-week administration period, at least 24 hours after the last nasal spray administration (T1); and at a follow-up session, four weeks after cessation of the daily administrations (T2). This follow-up timing was similar to a prior chronic oxytocin administration study showing elevated oxytocin levels after a 4-week follow-up in autistic adults[26].

Saliva samples were collected as exploratory, secondary endpoints in the context of a larger protocol, including clinical-behavioural[12] and neural[45,46] assessments as primary outcomes, as registered at EudraCT 2018-000769-35.

### Participants

Children with a formal diagnosis of autism were recruited through the Leuven Autism Expertise Centre of the between July 2019 and January 2021 (see Fig. 1 for the CONSORT flow diagram visualizing the number of included participants). All (travel) costs associated with the experimental procedures were reimbursed and children were given a toy of their choosing at the end of each assessment session. Participants randomized to receive oxytocin or placebo nasal sprays did not differ in terms of baseline symptom characteristics (Autism Diagnostic Observation Schedule, ADOS-2[47]; Social Responsiveness Scale-Children, SRS-2[48]), intelligence quotients (IQ; WISC-V-NL[49]), biological sex or age, see Table 1.

A total of 80 participants (40 in each treatment arm) participated in the trial, allowing to detect a medium effect size (d = 0.60) with α = 0.05 and 80% power, corresponding to effect sizes previously reported in a four-week oxytocin trial with school-aged children. Main inclusion criteria comprised a clinical diagnosis of ASD, age (8-12 years old), IQ above 70, no prior oxytocin administrations and native Dutch speaker. The ASD diagnosis was established by a multidisciplinary team (child psychiatrist, psychologist, speech/language pathologist, physiotherapist) based on the criteria of the Diagnostic and Statistical Manual of Mental Disorders, 5th edition[1].

Main exclusion criteria comprised a history of any neurological disorder (stroke, concussion, epilepsy etc.), any physical disorder

(liver, renal, cardiac pathology) or significant hearing or vision impairments. Only premenarchal girls were included. Seventeen participants had a co-occurring diagnosis of attention-deficit/hyperactivity disorder and one of obsessive-compulsive disorder. Twenty participants used stimulant medication (e.g., methylphenidate), 13 anti-psychotics (e.g. risperidone), 4 anti-depressants (e.g., sertraline) and 35 used other medication (e.g., sleeping aids, gastro-intestinal medication or nutritional supplements).

## Nasal spray administration

Participants received oxytocin (Syntocinon®, Sigma-tau) or placebo nasal sprays for four weeks (28 days), administered twice daily with 12 IU (3 puffs of 2 IU in each nostril) in the morning and 12 IU in the afternoon (after school), resulting in a daily dose of 24 IU, in accordance with prior studies assessing chronic oxytocin administration effects in children[50,51] and adults with autism[9].

Using permuted-block randomization, participants were randomly assigned to receive oxytocin or placebo nasal sprays, administered in identical blinded amber 10 ml glass bottles with a white nasal pump (0.05 ml or 2 IU /puff). Preparation, packaging and blinding of the nasal sprays was performed by Heidelberg University Hospital, Germany. Placebo sprays contained all the ingredients used in the active solution except the oxytocin compound. Participants were randomly assigned in a 1:1 ratio, with age, IQ and biological sex balanced across treatment arms (see Table 1). Children and their parents received clear instructions on how to administer the nasal spray[52] through a demonstration together with the experimenter. Two daily doses of 12 IU oxytocin nasal spray or placebo equivalent (3 puffs of 2 IU in each nostril) were administered: 12 IU in the morning and 12 IU in the afternoon, during 28 consecutive days. The last day before the post-nasal spray administration assessment session (T1), participants withheld the afternoon spray, to allow a window of 24 hours between the last nasal spray and the salivary sampling, thereby mitigating the potential for acute (single-dose) oxytocin effects. All research staff conducting the trial, participants and their parents were blinded to nasal spray allocation.

Compliance was monitored by weighing the disposed and returned nasal sprays, no differences were reported in the amount of administered fluid between the oxytocin and placebo group[12]. Potential adverse events were tracked using weekly parent reports and daily parent and child diaries. These revealed no evidence of nasal spray-specific adverse events (for a full report on compliance and adverse events see Daniels et al., (2023)[12]).

## Assessment of salivary oxytocin levels

Oxytocin levels were assessed via saliva samples acquired at each assessment session (T0, T1, T2), at two-time points: (i) a morning sample, acquired at home, within 30 min after awakening and before breakfast; and (ii) an afternoon sample, acquired at the Leuven University hospital. Salivary samples were collected using Salivette cotton swabs (Sarstedt AG & Co., Germany) and analysed using a commercial enzyme immunoassay oxytocin ELISA kit (Enzo Life Sciences, Inc., USA) in accordance with the manufacturer's instructions. Sample concentrations (100 μl/well) were calculated conform plate-specific standard curves.

## Assessment of *OXTR* DNA methylation levels

To assess variations in DNAm of *OXTR* (hg19, chr3:8,810,729-8,810,845), salivary samples were obtained via the Oragene DNA sample collection kit (DNA Genotek Inc., Canada) at the Leuven University hospital at T0, T1, and T2. Next, extracted DNA (200 ng) was bisulfite-converted following the manufacturer's protocol (EZ-96 DNAm Kit, Zymo Research, Irvine, USA) and stored at −80 °C. DNAm levels were determined at three CpG sites (−934, −924 and −914, relative to the translation start site) using Pyrosequencing (Qiagen, Germany) and analysed using Pyromark Q96 software, in accordance with manufacturer's protocols[53]. Protocols for the PCR amplification

and Pyrosequencing analysis were adopted from[54]. To amplify the DNA fragment of interest, within the *OXTR*, including the CpG sites -934, −924 and -914, the following PCR primers were used: [*OXTR* Forward: TTG AGT TTT GGA TTT AGA TAA TTA AGG ATT; *OXTR* Reverse: /5Biosg/AC TTA ACA TCA CAT TAA ATA CAA CC]. The following PCR amplification steps were used: Step 1: (95 °C/15 min)/1 cycle, Step 2: (94 °C/30 s, 58 °C/30 s, 72 °C/30 s)/50 cycles, Step 3: (72 °C/10 min)/1 cycle, Step 4: 4 °C hold]. The following sequencing primer was used to read the DNA sequence (*OXTR* Sequencing: AGA AGT TAT TTT ATA ATT TT).

## Data handling and statistical procedures

**Salivary oxytocin samples.** From oxytocin saliva samples containing less than 100 μl of material, 50 μl was obtained and diluted to 100 μl, sample concentrations were then multiplied by 2 (T0: 4 AM samples, 1 PM sample; T1: 3 AM samples; T2: 3 AM samples). Further, for participants displaying concentrations below the detection limit, concentrations were set to the lowest detected value across samples (T0: 14 AM samples, 12 PM samples; T1: 14 AM samples, 9 PM samples; T2: 18 AM samples, 9 PM samples). Finally, extreme outliers in oxytocin level data were identified (five inter-quartile ranges (Q3-Q1) below or above the first (Q1), respectively third (Q3) quartile) and recoded to the sample mean (across groups) (T0: 1 AM sample, 1 PM sample; T1: 3 AM samples; T2: 1 AM sample, 1 PM sample). Note that the pattern of results remained qualitatively similar also when removing these outliers.

To assess oxytocin-induced effects, salivary oxytocin levels were subjected to mixed-effects analyses of variances and Bonferroni-corrected post-hoc tests, with the factors '*nasal spray*' (oxytocin, placebo) and '*assessment session*' (T1, T2) as fixed effects, and the factor '*subject*' as random effect. To correct for variance in the individuals' baseline T0 scores, baseline values prior to nasal administration were included as a covariate in the model. Separate models were constructed for the morning and afternoon samples.

**Salivary DNAm samples.** Samples where *OXTR* DNAm could not be determined due to insufficient sample quality, were removed from analysis (T0: 2 samples, T1: 2 samples). For the DNAm level data, only one extreme outlier was identified and recoded to the sample mean (across groups), namely for CpG site -934, at T0.

Oxytocin-induced effects in *OXTR* DNAm were analysed using similar mixed models, i.e., with the fixed factors '*nasal spray*' (oxytocin, placebo) and '*assessment session*' (T1, T2), the random factor '*subject*', and baseline (T0) scores modelled as covariate. Separate models were constructed for each CpG site (-934, −924 and -914).

Next, Spearman correlation analyses were performed to examine whether variations in treatment-induced increases in salivary oxytocin levels (averaged across morning and afternoon), were associated to variations in treatment-induced changes in *OXTR* DNAm, as assessed in the oxytocin group. Considering that treatment-related effects on salivary oxytocin levels were only present at assessment session T1 (not at T2, see results), association analyses were primarily focused on this assessment session. For completeness, associations are also reported for follow-up session T2, as well as for T0, allowing to examine whether baseline variations in oxytocin levels possibly relate to baseline variations in *OXTR* DNAm.

Further, since prior work has evidenced a link between variations in oxytocin levels and behaviour[14,55], associations between salivary oxytocin levels and clinical-behavioural assessments were also performed. Specifically, exploratory examinations were performed to investigate whether individual variation in oxytocin-induced effects in endogenous oxytocinergic function relate to clinical-behavioural variations on social functioning (SRS-2[48]), repetitive behaviours (Repetitive Behavior Scale-Revised, RBS-R[56]), anxiety scores (Screen for Child Anxiety Related Emotional Disorders, SCARED-NL[57]), and self-reported attachment (Attachment Style Classification Questionnaire, ASCQ[58])

(see Table S1 for a detailed outlining of the obtained questionnaires). All statistical analyses were executed with Statistica version 14 (Tibco Software Inc.).

## Reporting summary

Further information on research design is available in the Nature Portfolio Reporting Summary linked to this article.

## Data availability

Data supporting the findings of the current report are available in Supplementary Data Source files. Source data are provided with this paper.

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

## Acknowledgements

This work was supported by a KU Leuven grant (C14/17/102), a Doctor Gustave Delport fund of the King Baudouin Foundation (2019-J1811190-212989) and a TBM grant of the Flanders Fund for Scientific Research (FWO-TBM T001821N) granted to K.A. and B.B., as well as by the Branco Weiss fellowship of the Society in Science – ETH Zurich granted to K.A. and the Excellence of Science grant (EOS; G0E8718N; HUMVISCAT) and Flanders Fund for Scientific Research grant (FWO; G023923N) granted to B.B.. M.M. is supported by a KU Leuven Postdoctoral Mandate. J.P. is supported by the Marguerite-Marie Delacroix foundation and a postdoctoral fellowship of the Flanders Fund for Scientific Research (FWO; 1257621 N). T.T. is supported by the Fund Child Hospital UZ Leuven. S.VdD. is supported by a KU Leuven Postdoctoral Mandate and a postdoctoral fellowship of the Flanders Fund for Scientific Research (FWO; 12C9723N). M.E. is supported by an aspirant fellowship of the Flanders Fund for Scientific Research (FWO; 11N1222N).

## Author contributions

M.M. conceptualization, methodology, investigation, data curation, validation, writing—original draft, writing—review and editing, visualization, project administration. N.D. conceptualization, investigation, data curation, validation, writing—review and editing, project administration. L.T. methodology, investigation, data curation, validation, writing—review and editing. T.T. methodology, investigation, data curation, validation, writing—review and editing. M.E. methodology, writing—review and editing. S.VdD. data curation, validation, writing—review and editing. E.D.: data curation, validation, writing—review and editing. J.P. data curation, validation, writing—review and editing. V.C. methodology, writing—review and editing. S.C. methodology, writing—review and editing. B.V. methodology, writing—review and editing. L.W. methodology, writing—review and editing. J.S. supervision, validation, writing—review and editing, funding acquisition. B.B. supervision, conceptualization, validation, writing—review and editing, funding acquisition. K.A. supervision, conceptualization, methodology, validation, writing—original draft, writing—review and editing, visualization, project administration, funding acquisition. All authors have read and agreed to the published version of the manuscript.

## Competing interests

The authors declare no competing interest.
