## [Peer Review File · Nature Communications]

Chronic oxytocin administration stimulates the endogenous oxytocin system in children with autismREVIEWER COMMENTS

Reviewer #1 (Remarks to the Author):

Summary and general points:

The authors explored the evidence how chronic administration of oxytocin impact on endogenous oxytocinergic function by analysing effects of four week intranasal oxytocin administration (12 IU, twice daily) on salivary oxytocin levels and DNA methylation of oxytocin receptor gene (OXTR) in 40 children with autism spectrum disorder (ASD) compared with four week placebo administration in 40 children with ASD in double-blind, randomized, placebo-controlled and parallel group design. Collection of saliva was performed at baseline (pre-treatment), 24 hours and four weeks after the last nasal spray administration of four week administration period. Compared with placebo, children receiving the oxytocin showed significantly higher salivary oxytocin levels 24 hours after the last oxytocin administration, but no longer at the four-week after that. Moreover, oxytocin-induced reductions in OXTR methylation were observed compared with placebo. The post-treatment increased salivary oxytocin levels were significantly correlated with the reduced OXTR DNA methylation and improved feelings of secure attachment. The authors interpreted the oxytocin-induced increased salivary oxytocin levels and reduced OXTR DNA methylation as that four weeks oxytocin administration stimulated the endogenous oxytocinergic system in children with ASD. The current study showed the novel, interesting, and relevant findings with robust statistical results. Although the findings are highly valuable for the progress of this research field, the current study is analyses of secondary endpoint and exploratory outcome measures of their clinical trial. In addition, stimulated endogenous oxytocinergic system or increased oxytocin receptor expression was not directly revealed by the current results. Therefore, some expressions of manuscript should be reconsidered from this viewpoint.

Specific points:

The authors should clarify the relationships between the previous and current papers on the same clinical trial from the authors group. Based on the registered information, the main outcomes of interest in current study is the secondary endpoint and exploratory outcome measures of the clinical trial. However, the current manuscript did not clarify the primary, the secondary, and the exploratory endpoints of this trial and the relationships between the findings from these endpoints.

The relationship between the measured salivary oxytocin level and endogenous oxytocin level was just inferred based on previous literature. The current study did not test whether the salivary oxytocin level reflect endogenous oxytocin level. Therefore, the expression such as "...explored chronic oxytocin administration effects on endogenous salivary oxytocin levels...(in abstract)" should be reconsidered in more appropriate manner.

Similarly, although previous study suggested that methylation of oxytocin receptor reflects oxytocin receptor expression, no direct evidence for the link between methylation of oxytocin receptor and oxytocin receptor expression in the current study. Therefore, the expression of manuscript should be reconsidered. For example, "OXTR methylation were observed, reflecting a facilitation of oxytocin receptor expression in the oxytocin..." in the abstract seems to be overexpression, and should be revised.

How to determine the timing of saliva collection such as 24 hours later or 4 week later the final administration? Brief explanation might help readers to understand the rationale.

Reviewer #2 (Remarks to the Author):

The manuscript by Moerkerke et al investigates the effects of chronic (4 week) intranasal oxytocin administration on endogenous oxytocin levels and oxytocin receptor methylation in autistic children. They found evidence for the oxytocin-induced oxytocin release hypothesis (positive feed-forward release) as the levels of salivary oxytocin were increased at 24hrs after the treatment was stopped. However, 4 weeks after treatment there was no more evidence of increased oxytocin levels. They also find that the methylation of the oxytocin receptor gene decreased after chronic administration of intranasal oxytocin, possibly indicating more oxytocin receptor expression. The authors found a correlation between behavioral assessment and oxytocin levels one day after the treatment was stopped.

Overall, I think the work is important and relevant in the context of the recent confusion in the scientific community regarding the role of oxytocin in behavior and its potential as a treatment for autism. The manuscript is clear and well-written.

The reviewer has several questions:

1. What is the relationship between behavior and oxytocin levels at T0 and T2? We would expect to see a correlation at those time points as well, right? And if all 3 time points are combined, perhaps the effect is stronger.
2. Similarly, are salivary oxytocin levels and receptor methylation correlated at T0 and T2?
3. In the discussion, could the authors address the specific epigenetic effect at site 924? Functionally, how is that different than methylation at the other 2 sites considered?
4. In the introduction, at line 53, there seems to be a missing word at the beginning of the sentence; "the past decade,..."

Reviewer #3 (Remarks to the Author):

This study investigated the impact of a four-week chronic oxytocin administration on endogenous oxytocinergic function in autistic children. After the last oxytocin nasal spray, salivary oxytocin levels increased, but this effect was no longer observed at the follow-up session four weeks after stopping the oxytocin administrations. The study also found reduced OXTR DNAm levels, suggesting increased expression of oxytocin receptors in the oxytocin group compared to the placebo group, even up to four weeks after the last nasal spray administration. The increase in oxytocin levels was associated with reduced OXTR DNAm levels and improved self-reported feelings of secure peer attachment. The study is important in contributing to the understanding of how oxytocin might impact individuals with ASD and the potential role of oxytocin methylation in mediating its effects. Investigating modifications in the OXTR gene can provide insights into the underlying biological mechanisms that regulate the oxytocin system and its effects on behavior and physiology.

The sample size and robust methodology are essential aspects of a study's strength. Importantly, it provides initial evidence for the potential of oxytocin as a treatment for social difficulties.

I have minor comments:

The introduction should include information regarding the main theories of how OT regulate social behavior (for example, the stress reduction theory and the social salience theory).

Also it is important to report previous attempt to use chronically intranasal oxytocin in ASD

More details are needed to explain the mechanisms of how chronic oxytocin administration affects endogenous oxytocin levels and gene expression.

The study suggests that combining oxytocin administration with social stimulation might perpetuate the endogenous oxytocin feedforward loop, but further research is needed to explore this. How exactly this could be achieved?

Please justify the dose administration (12IU twice daily)

The paper may better address potential confounding variables that could influence the outcomes, such as the participants' age, sex, or baseline oxytocin levels.

How did the author decide about the length of the Follow-up Period

Lack of Behavioral Outcomes: The study primarily focuses on changes in oxytocin levels and receptor expression but does not investigate real-life improvements in social functioning. What was the rationale behind the selection of the specific behavioral measures.

It is recommended to justify the focus on school-aged boys with autism, which limits the generalizability of the findings to other populations, including girls and adults with autism.

It is important to describe in more details the potential side effects or adverse reactions associated with chronic oxytocin administration in autistic children.

Response letter NCOMMS-23-24767

We appreciate the editors and reviewers careful reading and positive appraisal of our manuscript.

Reviewers comments are marked in blue, our responses in black. Revisions in the manuscript are highlighted in yellow.

REVIEWER COMMENTS

Reviewer #1 (Remarks to the Author):

Thank you for the thorough review and valuable feedback. We have tried to answer your questions to the best of our ability and rework the manuscript as adequately as possible.

Summary and general points: The authors explored the evidence how chronic administration of oxytocin impact on endogenous oxytocinergic function by analysing effects of four week intranasal oxytocin administration (12 IU, twice daily) on salivary oxytocin levels and DNA methylation of oxytocin receptor gene (OXTR) in 40 children with autism spectrum disorder (ASD) compared with four week placebo administration in 40 children with ASD in double-blind, randomized, placebo-controlled and parallel group design. Collection of saliva was performed at baseline (pre-treatment), 24 hours and four weeks after the last nasal spray administration of four week administration period. Compared with placebo, children receiving the oxytocin showed significantly higher salivary oxytocin levels 24 hours after the last oxytocin administration, but no longer at the four-week after that. Moreover, oxytocin-induced reductions in OXTR methylation were observed compared with placebo. The post-treatment increased salivary oxytocin levels were significantly correlated with the reduced OXTR DNA methylation and improved feelings of secure attachment. The authors interpreted the oxytocin-induced increased salivary oxytocin levels and reduced OXTR DNA methylation as that four weeks oxytocin administration stimulated the endogenous oxytocinergic system in children with ASD. The current study showed the novel, interesting, and relevant findings with robust statistical results. Although the findings are highly valuable for the progress of this research field, the current study is analyses of secondary endpoint and exploratory outcome measures of their clinical trial. In addition, stimulated endogenous oxytocinergic system or increased oxytocin receptor expression was not directly revealed by the current results. Therefore, some expressions of manuscript should be reconsidered from this viewpoint.

Specific points:

1. The authors should clarify the relationships between the previous and current papers on the same clinical trial from the authors group. Based on the registered information, the main outcomes of interest in current study is the secondary endpoint and exploratory outcome measures of the clinical trial. However, the current manuscript did not clarify the primary, the secondary, and the exploratory endpoints of this trial and the relationships between the findings from theses endpoints.

As correctly inferred by the reviewer, the collection of the saliva samples was indeed included as an exploratory, secondary objective in the larger protocol, as stated in the EudraCT registration. We have now included a paragraph at the end of the introduction section, explaining to the reader that the salivary sampling was part of a larger protocol, which also included the assessment of oxytocin-induced changes in distinct clinical-behavioural scales. We also state more explicitly the aim to conduct exploratory examinations of possible relationships between findings from these endpoints.

In the revised manuscript, we have also detailed this information in the methods section and highlighted the primary outcome measures, citing the corresponding publications for reference.

Added in the introduction, page 4:

“The current study aims to fill this gap, exploring the biological effects of a four-week course of chronic oxytocin administration on oxytocinergic function as assessed using salivary oxytocin levels and *OXTR* DNAm in school-aged children with ASD. Importantly, the exploratory biological salivary collections were part of a larger protocol, additionally including the assessment of oxytocin-induced changes on clinical-behavioural scales, as outlined in more detail in Daniels et al. (2023). Overall, Daniels et al. showed that -compared to children receiving placebo- children receiving oxytocin did not display stronger improvements in parent-reported social functioning (assessed as primary behavioural outcome using the social responsiveness scale) or repetitive behaviours, anxiety or attachment, even though the combination of oxytocin administration with psychosocial treatment might suggest a synergistic effect (Daniels et al., 2023). To understand possible inter-individual variations in clinical treatment outcomes, in the present report we explore whether individual variation in oxytocin-induced effects on endogenous oxytocinergic function may relate to variation in clinical effects.”

Added in the methods section, page 4:

“Saliva samples were collected as exploratory, secondary endpoints in the context of a larger protocol, which included clinical-behavioural (Daniels et al., 2023) and neural (Alaerts et al., 2023; Moerkerke et al., 2023) assessments as primary outcomes, as registered at the European Clinical Trial Registry (EudraCT 2018-000769-35) and the Belgian Federal Agency for Medicines and Health products (see Supplementary methods).”

2. The relationship between the measured salivary oxytocin level and endogenous oxytocin level was just inferred based on previous literature. The current study did not test whether the salivary oxytocin level reflect endogenous oxytocin level. Therefore, the expression such as “...explored chronic oxytocin administration effects on endogenous salivary oxytocin levels...(in abstract)” should be reconsidered in more appropriate manner.

Excellent point, we are indeed not able to directly discriminate between exogenous and endogenous oxytocin. However, peripheral oxytocin has a half-life of several minutes (as it is broken down by peptidases in the gastro-intestinal track and by oxytocinase in the blood). Based on this knowledge, we can infer that the oxytocin levels measured >24 hours after the last exogenous administration, could only originate from endogenously produced oxytocin.

Importantly, as we did not collect *central* oxytocin samples (from the cerebrospinal fluid – too invasive for young children), we are indeed unable to make statements on brain-oxytocin levels. In order to avoid too speculative phrasing, we adapted the manuscript, according to the reviewer’s suggestion.

Adapted in the abstract:

“...explored chronic oxytocin administration effects on endogenous peripheral, salivary oxytocin levels...”

Adapted in the discussion, page 8:

“The stimulating effect of chronic oxytocin administration on endogenous peripheral oxytocin levels in autistic children...”

Adapted in the discussion, page 9:

“...four weeks of chronic oxytocin administration was shown to stimulate the endogenous oxytocinergic system...”

3. Similarly, although previous study suggested that methylation of oxytocin receptor reflects oxytocin receptor expression, no direct evidence for the link between methylation of oxytocin receptor and oxytocin receptor expression in the current study. Therefore, the expression of manuscript should be reconsidered. For example, "OXTR methylation were observed, reflecting a facilitation of oxytocin receptor expression in the oxytocin..." in the abstract seems to be overexpression, and should be revised.

Thank you also for this valid consideration. While previous research has established a clear association between gene methylation and the suppression of gene expression (Gregory et al., 2009; Kusui et al., 2001), the current study did not include an explicit evaluation of *OXTR* expression levels (e.g. via mRNA levels), so no direct link can be established here. As suggested, we have now reduced the emphasis on this association throughout the manuscript, and also included a sentence in the 'limitations/future directions' paragraph, stating that future research is warranted to further examine the direct association between altered methylation at CpG site -924 and gene expression.

Added in the limitations part of the discussion, page 9:

"Moreover, the *OXTR* DNAm results were only found at CpG site -924, even though all three investigated sites fall within the same region in the CpG island of the *OXTR*, shown to regulate the expression of the oxytocin receptor (Kusui et al., 2001). Future studies should aim to clarify this differential effect, and specifically, how altered methylation at CpG site -924 might have impacted oxytocin receptor gene expression."

Adapted in the abstract:

"reductions in *OXTR* methylation were observed, suggesting a facilitation of oxytocin receptor expression in the oxytocin..."

"reduced *OXTR* DNA methylation (suggesting increased receptor expression)."

Adapted in the discussion, page 8:

"...suggesting a facilitation in oxytocin-receptor expression, as previously evidenced (Gregory et al., 2009; Kusui et al., 2001), ..."

Adapted in the discussion, page 9:

"...decreased *OXTR* DNAm levels (previously shown to reflect increased oxytocin-receptor expression (Gregory et al., 2009; Kusui et al., 2001))."

4. How to determine the timing of saliva collection such as 24 hours later or 4 week later the final administration? Brief explanation might help readers to understand the rationale.

Thank you for allowing us to elaborate on this in the manuscript. As previously noted, oxytocin has a short half-life of just a few minutes. However, several single-dose studies have reported elevated oxytocin levels persisting from 2 to 7 hours after administration (mentioned in the introduction) (Daughters et al., 2015; Huffmeijer et al., 2012; Procyshyn et al., 2020; Quintana et al., 2018; Riem et al., 2019; van IJzendoorn et al., 2012; Weisman et al., 2012). In the present manuscript, our primary aim was to investigate the persisting effects of chronic oxytocin administration, and thus maximally avoid any acute (single-dose) oxytocin effects or measure residual exogenous oxytocin. Consequently, we chose to cease administration at least one day (24 hours) prior to the post-treatment test session.

The length of the four-week follow-up period was based on the chronic oxytocin administration study in autistic adults of Alaerts et al. (2021), showing elevated salivary oxytocin levels 24 hours and four weeks after cessation of the nasal spray administration period.

Added in the methods section, page 4:

“On day 28, i.e. the last day before the post-nasal spray administration assessment (T1), participants withheld the afternoon spray, to allow a window of 24 hours between the last nasal spray and the salivary sampling, thereby mitigating the potential for acute (single-dose) oxytocin effects.”

“...at a follow-up session, four weeks after cessation of the daily administrations (T2). This follow-up timing was similar to a prior chronic oxytocin administration study showing elevated oxytocin levels after a four-week follow-up in autistic adults (Alaerts et al., 2021)”

Reviewer #2 (Remarks to the Author):

Thank you very much for the highly relevant remarks, we hope to have addressed them appropriately.

The manuscript by Moerkerke et al investigates the effects of chronic (4 week) intranasal oxytocin administration on endogenous oxytocin levels and oxytocin receptor methylation in autistic children. They found evidence for the oxytocin-induced oxytocin release hypothesis (positive feed-forward release) as the levels of salivary oxytocin were increased at 24hrs after the treatment was stopped. However, 4 weeks after treatment there was no more evidence of increased oxytocin levels. They also find that the methylation of the oxytocin receptor gene decreased after chronic administration of intranasal oxytocin, possibly indicating more oxytocin receptor expression. The authors found a correlation between behavioral assessment and oxytocin levels one day after the treatment was stopped.

Overall, I think the work is important and relevant in the context of the recent confusion in the scientific community regarding the role of oxytocin in behavior and its potential as a treatment for autism. The manuscript is clear and well-written.

The reviewer has several questions:

1. What is the relationship between behavior and oxytocin levels at T0 and T2? We would expect to see a correlation at those time points as well, right? And if all 3 time points are combined, perhaps the effect is stronger. Similarly, are salivary oxytocin levels and receptor methylation correlated at T0 and T2?

Thank you for this very relevant suggestion.

In our report, we primarily focused on the analyses of assessment session T1, because the strongest effect of the chronic oxytocin administration on salivary oxytocin levels was evidenced at this time point. We now describe this rationale more explicitly in the methods and results section. However, we do agree that it may be relevant to also include a description of possible relationships at follow-up T2 (as already briefly done), but also at T0, allowing to examine whether baseline variations in oxytocin levels possibly relate to baseline variations in *OXTR* DNAm.

In brief, and as now also incorporated in the manuscript, correlation analysis at T2 or T0 did not reveal significant relationships between oxytocin levels and DNA methylation or behaviour (all $p > 0.17$).

The related changes are now included as follows in the methods and results section.

Added to the methods section, page 6:

Next, Spearman correlation analyses were performed to examine whether variations in treatment-induced increases in salivary oxytocin levels (averaged across morning and afternoon), were associated to variations in treatment-induced changes in *OXTR* DNAm, as assessed in the oxytocin group. Considering that treatment-related effects on salivary oxytocin levels were only present at assessment session T1 (not at T2, see results), association analyses were primarily focused on this assessment session. For completeness, associations are also reported for follow-up session T2, as well as for T0,

allowing to examine whether baseline variations in oxytocin levels possibly relate to baseline variations in *OXTR* DNAm.

Further, since prior work has evidenced a link between variations in oxytocin levels and behaviour (Alaerts et al., 2019; Moerkerke, Peeters, et al., 2021), associations between salivary oxytocin levels and clinical-behavioural assessments were also performed. Specifically, exploratory examinations were performed to investigate whether individual variation in oxytocin-induced effects in endogenous oxytocinergic function relate to clinical-behavioural variations on social functioning (SRS-2, Constantino & Gruber, 2012), repetitive behaviours (Repetitive Behavior Scale-Revised, RBS-R, Bodfish et al., 2000), anxiety scores (Screen for Child Anxiety Related Emotional Disorders, SCARED-NL, Muris et al., 2007), and self-reported attachment (Attachment Style Classification Questionnaire, ASCQ, Finzi et al., 2000) (see **table S2** for a detailed outlining of the obtained questionnaires). All statistical analyses were executed with Statistica version 14 (Tibco Software Inc.). “

Added to the results section, page 7-8:

“Note that the relationship was only evident at the T1 assessment session - at which treatment-induced elevations in salivary oxytocin levels were observed – but no longer at the T2 follow-up session ($\rho = -.21$; $p = .214$). Also, no significant association was evident at the T0 session ($\rho = .15$; $p = .371$), indicating no significant relationship between baseline oxytocin levels and baseline DNAm of CpG site -924.”

“A significant relationship was identified between salivary oxytocin levels assessed from the oxytocin group at T1 (at which treatment-induced elevations were evident) and self-reports of secure attachment towards peers (ASCQ), indicating that children of the oxytocin group displaying higher oxytocin levels at T1 also showed higher reports of secure attachment at T1 ($\rho = .35$; $p = .030$), see **Fig. 3A**. Moderate (but non-significant) opposite relationships were evident for self-reported attachment avoidance ($\rho = -.30$; $p = .070$, **Fig. 3B**) and parent-reports of social difficulties (SRS-2; $\rho = -.31$; $p = .062$, **Fig. 3C**) and anxiety (SCARED-NL; $\rho = -.32$; $p = .053$, **Fig. 3D**) all indicating that improved clinical presentations at T1 were associated with higher T1 oxytocin levels. No significant associations were evident between oxytocin levels and clinical-behavioural scales at the follow-up session T2 (all $p > .208$), or at the baseline T0 assessment (all $p > .170$).

Also no significant associations were identified between levels of DNAm of CpG site -924 and the abovementioned clinical scales, either at T1, T2 or T0 (all $p > .108$).”

2. In the discussion, could the authors address the specific epigenetic effect at site 924? Functionally, how is that different than methylation at the other 2 sites considered?

This is a very valid question. The included CpG sites (-934, -924 and -914, relative to the translation start site) fall within the critical MT2 region in the CpG island of *OXTR*, shown to regulate the expression of the oxytocin receptor. Note that altered methylation at sites -934 and -924 has previously been implicated in autistic and altered social behaviour (Moerkerke et al., 2021). In principle, an epigenetic effect could be expected to occur on all three sites, thus future studies might be warranted to further examine this differential effect, and specifically, how the specific changes at CpG sites -924 might have impacted oxytocin receptor expression (see also comment 3 from Reviewer 1).

We now have addressed this issue in the limitations/future directions part of the discussion.

Added to the methods section, page 5:

“(i.e., -934, -924 and -914, relative to the translation start site; see (Moerkerke, Bonte, et al., 2021))”

Added in the limitations part of the discussion, page 9:

“ Moreover, the *OXTR* DNAm results were only found at CpG site -924, even though all three investigated sites fall within the same region in the CpG island of the *OXTR*, shown to regulate the expression of the oxytocin receptor (Kusui et al., 2001). Future studies should aim to clarify this differential effect, and specifically, how altered methylation at CpG site -924 might have impacted oxytocin receptor gene expression.”

3. In the introduction, at line 53, there seems to be a missing word at the beginning of the sentence; “the past decade,…”

Thank you very much for noticing this mistake.

Adapted in the introduction, page 3:

“In the past decade, intranasal administration of oxytocin **has been** increasingly explored as …”

Reviewer #3 (Remarks to the Author):

This study investigated the impact of a four-week chronic oxytocin administration on endogenous oxytocinergic function in autistic children. After the last oxytocin nasal spray, salivary oxytocin levels increased, but this effect was no longer observed at the follow-up session four weeks after stopping the oxytocin administrations. The study also found reduced *OXTR* DNAm levels, suggesting increased expression of oxytocin receptors in the oxytocin group compared to the placebo group, even up to four weeks after the last nasal spray administration. The increase in oxytocin levels was associated with reduced *OXTR* DNAm levels and improved self-reported feelings of secure peer attachment. The study is important in contributing to the understanding of how oxytocin might impact individuals with ASD and the potential role of oxytocin methylation in mediating its effects. Investigating modifications in the *OXTR* gene can provide insights into the underlying biological mechanisms that regulate the oxytocin system and its effects on behavior and physiology. The sample size and robust methodology are essential aspects of a study's strength. Importantly, it provides initial evidence for the potential of oxytocin as a treatment for social difficulties.

I have minor comments:

1. The introduction should include information regarding the main theories of how OT regulate social behavior (for example, the stress reduction theory and the social salience theory). Also it is important to report previous attempt to use chronically intranasal oxytocin in ASD

Thank you for allowing us to elaborate more on the oxytocin theories. We have now added this information in the introduction, page 3:

“To date, two main mechanistic frameworks on oxytocin's role in regulating social behaviour have been proposed. First, the anxiolytic account suggests that oxytocin primarily regulates stress and social anxiety responses, thereby promoting social approach behaviour during interactions (Bethlehem et al., 2014; Ma et al., 2016; Quintana et al., 2015; Stoop, 2014). Secondly, the social salience hypothesis proposes that oxytocin enhances attention to and perception of social cues by prioritizing neural resources for processing these cues (Shamay-Tsoory & Abu-Akel, 2016).”

Regarding the reporting of information on previous chronic intranasal oxytocin studies in ASD, we now also included for completeness the reference to the work of Horta et al. (2020), reporting on a full overview of multiple-dose, chronic oxytocin trials in ASD (and other populations).

Please find these associated changes in the introduction on, page 3:

“...multiple-dose, chronic administration studies (i.e. administering the oxytocin nasal spray over a course of multiple weeks) have yielded a more mixed pattern of results, with some studies demonstrating beneficial outcomes, while others did not (Anagnostou et al., 2012; Bernaerts et al., 2020; Daniels et al., 2023; Le et al., 2022; Sikich et al., 2021; see Horta et al., 2020, for a full review on chronic oxytocin administration trials).

2. More details are needed to explain the mechanisms of how chronic oxytocin administration affects endogenous oxytocin levels and gene expression.

Thank you for this suggestion.

While the exact mechanisms are not fully elucidated, we now added a bit more detail regarding the purported mechanism by which oxytocin administration may impact its own feedforward release.

We now provided additional information in the discussion, page 8:

“Particularly, an auto-excitatory mechanism of somato-dendritic release of oxytocin from hypothalamic cells has been described, indicating that auto-binding of oxytocin to its G protein-coupled receptors is sufficient to elevate intracellular Ca^{2+} levels from internal storage, in turn triggering a feed-forward of dendritic oxytocin exocytosis (i.e. release) (Jurek & Neumann, 2018; Ludwig et al., 2002).

With regard to the regulation of *OXTR* expression and receptor availability, prior rodent research showed mixed results, with some studies showing a receptor downregulation upon chronic application of oxytocin in mice (10 ng/h, over a period of 14 days) (Huang et al., 2013; Peters et al., 2014), while others showed opposite effects (Zimmermann-Peruzatto et al., 2017). For example, in oxytocin knockout mice, the expression of the *OXTR* gene was shown to be reduced in hippocampal tissue, due to unavailability of the ligand (Zimmermann-Peruzatto et al., 2017). In the current human sample, the chronic oxytocin administration induced a predominant decrease in *OXTR* methylation, previously shown to be reflective of increased receptor expression (Gregory et al., 2009; Kusui et al., 2001). It is likely however, that the effects on gene regulation of (chronic) oxytocin administration are dependent on the administered dose and frequency, with some dosing schemes inducing facilitation, while others might induce downregulation of receptor availability. In a recent study by Le et al. (2022), the possibility of an intermittent dosing scheme - administering oxytocin intranasally every other day for a period of 6 weeks to avoid pharmacological overstimulation - was explored in a paediatric sample of young autistic children, and although no direct assessments of epigenetic changes were reported, the administration regime was shown to yield strong beneficial effects on various clinical-behavioural outcomes assessing social function (Le et al., 2022). It should be interesting for future studies to further explore optimal dosing schema's for yielding maximal endogenous stimulation of the oxytocinergic system.”

3. The study suggests that combining oxytocin administration with social stimulation might perpetuate the endogenous oxytocin feedforward loop, but further research is needed to explore this. How exactly this could be achieved?

Thank you for this comment. We now added an extra sentence to the discussion to elaborate a bit more on this notion of combinatory approaches for oxytocin administration. .

Please find the associated changes in the discussion, page 8:

“...this highlights the importance of combining oxytocin administration with social stimulation (e.g. psychosocial therapy) to perpetuate the endogenous oxytocin feedforward-loop (De Dreu, 2012). This aligns with emerging evidence indicating that oxytocin administration as stand-alone treatment may be suboptimal (Ford & Young, 2021; Itskovich et al., 2022), and that combinatory approaches, combining the chronic oxytocin administrations to concomitant psychosocial training interventions will be pivotal for maximizing its full therapeutic potential and clinical efficacy (Daniels et al., 2023; Le et al., 2022).”

4. Please justify the dose administration (12IU twice daily)

The current study adopted a total daily dose of 24 IU, administered as 12 IU in the morning and 12 IU in the afternoon, for a duration of four weeks. This dosing schema was chosen to be similar to the dosing adopted in the paediatric trial by Yatawara et al. (2015), i.e. adopting a similar daily dose of 24 IU, administered as 12 IU in the morning and 12 IU at night. The total daily dose of 24 IU also aligns with other prior studies, including e.g. Dadds et al. (2014) and Bernaerts et al. (2020).

Added in the methods section, page 5:

“Participants received oxytocin (Syntocinon®, Sigma-tau) or placebo nasal sprays for four weeks, administered twice daily with six puffs (three per nostril) of 12 IU in the morning and six puffs of 12 IU in the afternoon (after school), resulting in a daily dose of 24 IU, in accordance with prior studies assessing chronic oxytocin administration effects in children (Dadds et al., 2014; Yatawara et al., 2016) and adults with autism (Bernaerts et al., 2020).”

5. How did the author decide about the length of the Follow-up Period

The four week follow-up period was chosen to be similar to the chronic oxytocin administration study previously conducted in autistic adults by Alaerts et al. (2021), showing elevated salivary oxytocin levels 24 hours and four weeks after cessation of the nasal spray administration period. This is briefly mentioned in the introduction:

“...in autistic adult men (Alaerts et al., 2021). Here, elevated salivary oxytocin levels were shown up to four weeks after cessation of the nasal spray...”

To emphasize our rationale for choosing the duration of the follow-up period, we now included this information in the methods section as well.

Added in the methods section, page 4:

“...at a follow-up session, four weeks after cessation of the daily administrations (T2). This follow-up timing was chosen to be similar to a prior chronic oxytocin administration study showing elevated oxytocin levels after a four week follow-up in autistic adults (Alaerts et al., 2021).”

6. Lack of Behavioral Outcomes: The study primarily focuses on changes in oxytocin levels and receptor expression but does not investigate real-life improvements in social functioning. What was the rationale behind the selection of the specific behavioral measures.

The collection of the saliva samples were part of a larger assessment protocol, evaluating clinical efficacy of oxytocin administration on distinct autism symptoms questionnaires, including the parent-rated Social Responsiveness Scale-Children, 2nd edition (SRS-2, commonly used in autism research) as primary behavioural outcome, as well as other scales assessing constructs of repetitive behaviours, anxiety and attachment.

Since the effects of the chronic oxytocin administration on these clinical-behavioural outcomes were previously reported in detail in Daniels et al. (2023), we here focussed in our current report on the biological samplings as well as on the exploration of possible associations between variations in biological effects and individual variation in clinical-behavioural improvements.

While we briefly outlined this rationale in the method section, we agree that this placing was perhaps suboptimal, and that it would be more informative to the reader to learn about these planned biological-behavioural association analyses already earlier on, in the introduction section.

To do so, we now included an extra paragraph in the introduction, briefly summarizing the behavioural effects as reported in Daniels et al. (2023), and also outlining the planned correlation analyses.

Please find the associated changes in the introduction on page 4:

“Importantly, the exploratory biological salivary collections were part of a larger protocol, additionally including the assessment of oxytocin-induced changes on clinical-behavioural scales, as outlined in more detail in Daniels et al. (2023). Overall, Daniels et al. showed that -compared to children receiving placebo- the group of children receiving oxytocin did not display stronger improvements in parent-reported social functioning (assessed as primary behavioural outcome using the social responsiveness scale) or repetitive behaviours, anxiety or attachment, although the combination of oxytocin administration with psychosocial treatment might suggest a synergetic effect (Daniels et al., 2023). To understand possible inter-individual variations in clinical treatment outcomes, in the present report we explore whether individual variation in oxytocin-induced effects on endogenous oxytocinergic function may relate to variation in clinical effects.

Added to the methods section, page 6:

“Further, since prior work has evidenced a link between variations in oxytocin levels and behaviour (e.g., Alaerts et al., 2019), associations between salivary oxytocin levels and obtained clinical-behavioural assessments were also performed. Specifically, exploratory examinations were performed to discern whether individual variation in oxytocin-induced effects in endogenous oxytocinergic function relate to clinical-behavioural variations on social functioning (SRS-2, Constantino & Gruber, 2012), repetitive behaviours (Repetitive Behavior Scale-Revised, RBS-R, Bodfish et al., 2000), anxiety scores (Screen for Child Anxiety Related Emotional Disorders, SCARED-NL, Muris et al., 2007), and self-reported attachment (Attachment Style Classification Questionnaire, ASCQ, Finzi et al., 2000) (see **table S2** for a detailed outlining of the obtained questionnaires). “

7. It is recommended to justify the focus on school-aged boys with autism, which limits the generalizability of the findings to other populations, including girls and adults with autism.

This is an excellent point. Although we included girls into our sample, the majority of the participants were still boys (80%, reflecting the gender ratio in the general population – see Table 1, outlining the demographics of our sample). We acknowledge the limited generalizability due to this gender ratio and narrow age-range, and as such, we have emphasized this in the limitations part of the discussion.

“Further, considering the tight age range of our school-aged children with ASD, and a predominant sampling from boys with ASD, it should be explored whether the observed effects will generalize across different age ranges and in samples with a larger representation of girls.”

8. It is important to describe in more details the potential side effects or adverse reactions associated with chronic oxytocin administration in autistic children.

Thank you for this valid point. The full report on side effects can be found in the primary clinical-behavioural outcome manuscript (Daniels et al., 2023). This information has now been included, together with a brief summary of the results concerning side effects.

Added in the participant part of the methods section, page 4:

“Detailed information regarding inclusion criteria, participant screening and side effects is provided in Supplementary methods.”

Added in the participant part of the Supplementary methods, page 2:

“During the nasal spray administration period, potential adverse events were assessed in participants through weekly parent reports, and changes in affect and arousal were recorded in daily diaries by both the child and parent. Overall, reports of side effects were minimal and not treatment-specific; for a comprehensive report, please consult Daniels et al., (2023).”

References

- Alaerts, K., Bernaerts, S., Vanaudenaerde, B., Daniels, N., & Wenderoth, N. (2019). Amygdala–Hippocampal Connectivity Is Associated With Endogenous Levels of Oxytocin and Can Be Altered by Exogenously Administered Oxytocin in Adults With Autism. *Biological Psychiatry: Cognitive Neuroscience and Neuroimaging*, *4*(7), 655–663. <https://doi.org/10.1016/j.bpsc.2019.01.008>
- Alaerts, K., Daniels, N., Moerkerke, M., Evenepoel, M., Tang, T., Van der Donck, S., Chubar, V., Stephan, C., Steyaert, J., Boets, B., & Prinsen, J. (2023). At the head and heart of oxytocin’s stress-regulatory neural and cardiac effects: a chronic administration RCT in children with autism. *MedRxiv*, 2023.04.04.23288109. <https://doi.org/10.1101/2023.04.04.23288109>
- Alaerts, K., Steyaert, J., Vanaudenaerde, B., Wenderoth, N., & Bernaerts, S. (2021). Changes in endogenous oxytocin levels after intranasal oxytocin treatment in adult men with autism: An exploratory study with long-term follow-up. *European Neuropsychopharmacology*, *43*, 147–152. <https://doi.org/10.1016/J.EURONEURO.2020.11.014>
- Anagnostou, E., Soorya, L., Chaplin, W., Bartz, J., Halpern, D., Wasserman, S., Wang, A. T., Pepa, L., Tanel, N., Kushki, A., & Hollander, E. (2012). Intranasal oxytocin versus placebo in the treatment of adults with autism spectrum disorders: A randomized controlled trial. *Molecular Autism*, *3*(1), 1–9. <https://doi.org/10.1186/2040-2392-3-16/TABLES/3>
- Bernaerts, S., Boets, B., Bosmans, G., Steyaert, J., & Alaerts, K. (2020). Behavioral effects of multiple-dose oxytocin treatment in autism: A randomized, placebo-controlled trial with long-term follow-up. *Molecular Autism*, *11*(1), 1–14. <https://doi.org/10.1186/s13229-020-0313-1>
- Bethlehem, R. A. I., Baron-Cohen, S., van Honk, J., Auyeung, B., & Bos, P. A. (2014). The oxytocin paradox. *Frontiers in Behavioral Neuroscience*, *8*(48), 5. <https://doi.org/10.3389/FNBEH.2014.00048/BIBTEX>
- Dadds, M. R., MacDonald, E., Cauchi, A., Williams, K., Levy, F., & Brennan, J. (2014). Nasal oxytocin for social deficits in childhood autism: A randomized controlled trial. *Journal of Autism and Developmental Disorders*, *44*(3), 521–531. <https://doi.org/10.1007/s10803-013-1899-3>
- Daniels, N., Moerkerke, M., Steyaert, J., Bamps, A., Debbaut, E., Prinsen, J., Tang, T., Van Der Donck, S., Boets, B., & Alaerts, K. (2023). Effects of multiple-dose intranasal oxytocin administration on social responsiveness in children with autism: a randomized, placebo-controlled trial. *Molecular Autism*, *14*, 16. <https://doi.org/10.1186/s13229-023-00546-5>

- Daughters, K., Manstead, A. S. R., Hubble, K., Rees, A., Thapar, A., & Van Goozen, S. H. M. (2015). *Salivary Oxytocin Concentrations in Males following Intranasal Administration of Oxytocin: A Double-Blind, Cross-Over Study*. <https://doi.org/10.1371/journal.pone.0145104>
- De Dreu, C. K. W. (2012). Oxytocin modulates cooperation within and competition between groups : An integrative review and research agenda. *Hormones and Behavior*, *61*(3), 419–428. <https://doi.org/10.1016/j.yhbeh.2011.12.009>
- Ford, C. L., & Young, L. J. (2021). Refining oxytocin therapy for autism: context is key. *Nature Reviews Neurology* *2021 18:2*, *18*(2), 67–68. <https://doi.org/10.1038/s41582-021-00602-9>
- Gregory, S. G., Connelly, J. J., Towers, A. J., Johnson, J., Biscocho, D., Markunas, C. A., Lintas, C., Abramson, R. K., Wright, H. H., Ellis, P., Langford, C. F., Worley, G., DeLong, G. R., Murphy, S. K., Cuccaro, M. L., Persico, A., & Pericak-Vance, M. A. (2009). Genomic and epigenetic evidence for oxytocin receptor deficiency in autism. *BMC Medicine*, *7*, 1–13. <https://doi.org/10.1186/1741-7015-7-62>
- Horta, M., Kaylor, K., Feifel, D., & Ebner, N. C. (2020). Chronic oxytocin administration as a tool for investigation and treatment: A cross-disciplinary systematic review. *Neuroscience and Biobehavioral Reviews*, *108*(April 2019), 1–23. <https://doi.org/10.1016/j.neubiorev.2019.10.012>
- Huang, H., Michetti, C., Busnelli, M., Managò, F., Sannino, S., Scheggia, D., Giancardo, L., Sona, D., Murino, V., Chini, B., Scattoni, M. L., & Papaleo, F. (2013). Chronic and Acute Intranasal Oxytocin Produce Divergent Social Effects in Mice. *Neuropsychopharmacology* *2014 39:5*, *39*(5), 1102–1114. <https://doi.org/10.1038/npp.2013.310>
- Huffmeijer, R., Alink, L. R. A., Tops, M., Grewen, K. M., Light, K. C., Bakermans-Kranenburg, M. J., & Van Ijzendoorn, M. H. (2012). Salivary levels of oxytocin remain elevated for more than two hours after intranasal oxytocin administration. *Neuroendocrinology Letters*, *33*(1), 22467107–330112. www.nel.edu
- Itskovich, E., Bowling, D. L., Garner, J. P., & Parker, K. J. (2022). Oxytocin and the social facilitation of placebo effects. *Molecular Psychiatry* *2022*, 1–10. <https://doi.org/10.1038/s41380-022-01515-9>
- Jurek, B., & Neumann, I. D. (2018). The oxytocin receptor: From intracellular signaling to behavior. *Physiological Reviews*, *98*(3), 1805–1908. <https://doi.org/10.1152/PHYSREV.00031.2017/ASSET/IMAGES/LARGE/Z9J0021828400013.JPEG>
- Kusui, C., Kimura, T., Ogita, K., Nakamura, H., Matsumura, Y., Koyama, M., Azuma, C., & Murata, Y. (2001). DNA methylation of the human oxytocin receptor gene promoter regulates tissue-specific gene suppression. *Biochemical and Biophysical Research Communications*, *289*(3), 681–686. <https://doi.org/10.1006/bbrc.2001.6024>
- Le, J., Zhang, L., Zhao, W., Zhu, S., Lan, C., Kou, J., Zhang, Q., Zhang, Y., Li, Q., Chen, Z., Fu, M., Montag, C., Zhang, R., Yang, W., Becker, B., & Kendrick, K. M. (2022). Infrequent Intranasal Oxytocin Followed by Positive Social Interaction Improves Symptoms in Autistic Children: A Pilot Randomized Clinical Trial. *Psychotherapy and Psychosomatics*, 1–13. <https://doi.org/10.1159/000524543>
- Ludwig, M., Sabatier, N., Bull, P. M., Landgraf, R., Dayanithi, G., & Leng, G. (2002). Intracellular calcium stores regulate activity-dependent neuropeptide release from dendrites. *Nature* *2002 418:6893*, *418*(6893), 85–89. <https://doi.org/10.1038/nature00822>
- Ma, Y., Shamay-Tsoory, S., Han, S., & Zink, C. F. (2016). Oxytocin and Social Adaptation: Insights from Neuroimaging Studies of Healthy and Clinical Populations. *Trends in Cognitive Sciences*, *20*(2), 133–145. <https://doi.org/10.1016/j.tics.2015.10.009>

- Moerkerke, M., Bonte, M. L., Daniels, N., Chubar, V., Alaerts, K., Steyaert, J., & Boets, B. (2021). Oxytocin receptor gene (OXTR) DNA methylation is associated with autism and related social traits – A systematic review. *Research in Autism Spectrum Disorders*, 85(April), 101785. <https://doi.org/10.1016/j.rasd.2021.101785>
- Moerkerke, M., Daniels, N., Donck, S. Van der, Tibermont, L., Tang, T., Debbaut, E., Bamps, A., Prinsen, J., Steyaert, J., Alaerts, K., & Boets, B. (2023). Can repeated intranasal oxytocin administration affect reduced neural sensitivity towards expressive faces in autism? A randomized controlled trial. *Journal of Child Psychology and Psychiatry*. <https://doi.org/10.1111/JCPP.13850>
- Moerkerke, M., Peeters, M., de Vries, L., Daniels, N., Steyaert, J., Alaerts, K., & Boets, B. (2021). Endogenous Oxytocin Levels in Autism—A Meta-Analysis. *Brain Sciences* 2021, Vol. 11, Page 1545, 11(11), 1545. <https://doi.org/10.3390/BRAINSCI11111545>
- Peters, S., Slattery, D. A., Uschold-Schmidt, N., Reber, S. O., & Neumann, I. D. (2014). Dose-dependent effects of chronic central infusion of oxytocin on anxiety, oxytocin receptor binding and stress-related parameters in mice. *Psychoneuroendocrinology*, 42, 225–236. <https://doi.org/10.1016/J.PSYNEUEN.2014.01.021>
- Procyshyn, T. L., Lombardo, M. V., Lai, M. C., Auyeung, B., Crockford, S. K., Deakin, J., Soubramanian, S., Sule, A., Baron-Cohen, S., & Bethlehem, R. A. I. (2020). Effects of oxytocin administration on salivary sex hormone levels in autistic and neurotypical women. *Molecular Autism*, 11(1), 1–11. <https://doi.org/10.1186/S13229-020-00326-5/FIGURES/3>
- Quintana, D. S., Westlye, L. T., Rustan, O. G., Tesli, N., Poppy, C. L., Smevik, H., Tesli, M., Røine, M., Mahmoud, R. A., Smerud, K. T., Djupesland, P. G., & Andreassen, O. A. (2015). Low-dose oxytocin delivered intranasally with Breath Powered device affects social-cognitive behavior: a randomized four-way crossover trial with nasal cavity dimension assessment. *Translational Psychiatry* 2015 5:7, 5(7), e602–e602. <https://doi.org/10.1038/tp.2015.93>
- Quintana, D. S., Westlye, L. T., Smerud, K. T., Mahmoud, R. A., Andreassen, O. A., & Djupesland, P. G. (2018). Saliva oxytocin measures do not reflect peripheral plasma concentrations after intranasal oxytocin administration in men. *Hormones and Behavior*, 102, 85–92. <https://doi.org/10.1016/J.YHBEH.2018.05.004>
- Riem, M. M. E., van IJzendoorn, M. H., & Bakermans-Kranenburg, M. J. (2019). Hippocampal volume modulates salivary oxytocin level increases after intranasal oxytocin administration. *Psychoneuroendocrinology*, 101, 182–185. <https://doi.org/10.1016/J.PSYNEUEN.2018.11.015>
- Shamay-Tsoory, S. G., & Abu-Akel, A. (2016). The Social Salience Hypothesis of Oxytocin. *Biological Psychiatry*, 79(3), 194–202. <https://doi.org/10.1016/j.biopsych.2015.07.020>
- Sikich, L., Kolevzon, A., King, B. H., McDougle, C. J., Sanders, K. B., Kim, S.-J., Spanos, M., Chandrasekhar, T., Trelles, M. D. P., Rockhill, C. M., Palumbo, M. L., Cundiff, A. W., Montgomery, A., Siper, P., Minjarez, M., Nowinski, L. A., Marler, S., Shuffrey, L. C., Alderman, C., ... Veenstra-VanderWeele, J. (2021). Intranasal Oxytocin in Children and Adolescents with Autism Spectrum Disorder. *The New Engl and Journal of Medicine*, 385(16), 1462–1473. <https://www.nejm.org/doi/10.1056/NEJMoa2103583>
- Stoop, R. (2014). Neuromodulation by oxytocin and vasopressin in the central nervous system as a basis for their rapid behavioral effects. *Current Opinion in Neurobiology*, 29, 187–193. <https://doi.org/10.1016/J.CONB.2014.09.012>

- van IJzendoorn, M. H., Bhandari, R., van der Veen, R., Grewen, K. M., & Bakermans-Kranenburg, M. J. (2012). Elevated Salivary Levels of Oxytocin Persist More than 7 h after Intranasal Administration. *Frontiers in Neuroscience*, *6*, 174. <https://doi.org/10.3389/FNINS.2012.00174>
- Weisman, O., Zagoory-Sharon, O., & Feldman, R. (2012). Intranasal oxytocin administration is reflected in human saliva. *Psychoneuroendocrinology*, *37*(9), 1582–1586. <https://doi.org/10.1016/j.psyneuen.2012.02.014>
- Yatawara, C. J., Einfeld, S. L., Hickie, I. B., Davenport, T. A., & Guastella, A. J. (2016). The effect of oxytocin nasal spray on social interaction deficits observed in young children with autism: A randomized clinical crossover trial. *Molecular Psychiatry*, *21*(9), 1225–1231. <https://doi.org/10.1038/mp.2015.162>
- Zimmermann-Peruzatto, J. M., Lazzari, V. M., Agnes, G., Becker, R. O., de Moura, A. C., Guedes, R. P., Lucion, A. B., Almeida, S., & Giovenardi, M. (2017). The Impact of Oxytocin Gene Knockout on Sexual Behavior and Gene Expression Related to Neuroendocrine Systems in the Brain of Female Mice. *Cellular and Molecular Neurobiology*, *37*(5), 803–815. <https://doi.org/10.1007/S10571-016-0419-3/FIGURES/6>

REVIEWERS' COMMENTS

Reviewer #1 (Remarks to the Author):

All the points raised with the initial review were perfectly improved by the revision.

Reviewer #2 (Remarks to the Author):

The authors have fully addressed my concerns in their revised submission.

Reviewer #3 (Remarks to the Author):

The authors addressed all my comments. I am happy with the revision

Response letter NCOMMS-23-24767A

We are grateful for the thorough review and favourable evaluation of our manuscript by the editors and reviewers. Reviewers' remarks are listed below, no further revision is needed.

Revisions made in the manuscript, based on the editors requests, are highlighted in **yellow**.

REVIEWER COMMENTS

Reviewer #1 (Remarks to the Author):

All the points raised with the initial review were perfectly improved by the revision.

Reviewer #2 (Remarks to the Author):

The authors have fully addressed my concerns in their revised submission.

Reviewer #3 (Remarks to the Author):

The authors addressed all my comments. I am happy with the revision.